# Systemic inflammatory markers of visceral leishmaniasis treatment response in East Africa

Ayenew Addisu[1]☉, Alice Bayiyana[2]☉, João L. Reis Cunha[3]☉, Daniel Matano[4]☉, Brima M. Younis[5]☉, Karen Hogg[6], Rebecca Wiggins[3], Wilson Biwott[4], Finnley Osuna[4], Christine Ichugu[4], Ayalew Jejaw Zeleke[1], Eleni Ayele[7], James Obondo Sande[2], Eltahir A. G. Khalil[5], Hussam M. H. Ibrahim[5], Mahmoud A. Mahmoud[5], Ahmed I. B. Zakaria[5], Brenda Adiko[2], Peter O'Toole[6], Flavia D'Alessio[8], Charles J. N. Lacey[3], Jane Mbui[4]*, Asrat M. Hailu[1]*, Paul M. Kaye🆔[3]*, Margaret Mbuchi[4]*, Ahmed M. Musa[5]*, Joseph Olobo[2]*, The Immstat@Cure Consortium[¶]

1 Department of Medical Parasitology, School of Biomedical and Laboratory Sciences, College of Medicine and Health Sciences, University of Gondar, Gondar, Ethiopia, 2 Department of Immunology and Molecular Biology, Makerere University, Kampala, Uganda, 3 York Biomedical Research Institute, Hull York Medical School, University of York, York, United Kingdom, 4 Center for Clinical Research, Kenya Medical Research Institute, Nairobi, Kenya, 5 Institute of Endemic Diseases, Khartoum, Sudan, 6 BioSciences Technology Facility, Dept of Biology, University of York, York, United Kingdom, 7 Leishmania Research and Treatment Center, University of Gondar Comprehensive and Specialized Hospital, Gondar, Ethiopia, 8 European Vaccine Initiative, Heidelberg, Germany

¶ Membership of The Immstat@cure consortium members are listed in the S1 Appendix.
☉ These authors contributed equally to the work.
* paul.kaye@york.ac.uk (PMK); hailu_a2004@yahoo.com (AMH); jmbui@kemri.go.ke (JM); Mmbuchi@kemri.go.ke (MM); musaam2003@yahoo.co.uk (AMM); oloboj@yahoo.co.uk (JO)

## Abstract

### Background

Visceral leishmaniasis (VL) is the most severe form of leishmaniasis, with East Africa accounting for ~70% of global burden. It primarily affects malnourished children, young adults, and HIV co-infected individuals. Clinical outcomes range from asymptomatic to fatal, with relapse mostly linked to HIV co-infection, splenomegaly, high parasite load, poor immune responses, and elevated IgG1 concentration. In rodent VL models, systemic immune and metabolic abnormalities persist at the end of the drug treatment regime. However, the immune status of VL patients in East Africa at the end of treatment is not fully understood.

### Methodology/principal findings

We conducted ImmStat@cure, a multicentre clinical study to assess clinical and immune profiles of VL patients at admission and end of treatment (EoT) in East African countries. Clinical, haematological and inflammatory markers data were collected from patients from Ethiopia, Kenya, Sudan and Uganda on both time points and from convenience controls at a single time point. By integrating clinical data with haematological and inflammation markers, we have shown that patient clinical and

**Data availability statement:** The R script used for data analysis can be obtained from GitHub: https://github.com/jaumlrc/Immstat_cure.git. Summaries and descriptions of average patient clinical and immunological data, as well as Individualized Anonymized patient data tables with patient clinical and immunological data can be seen in S1 Table.

**Funding:** This project is part of the EDCTP2 programme supported by the European Union (grant number RIA2016V-1640-PREV_PKDL). JC, RW and PMK were also supported by a Wellcome Trust Investigator Award (#224290 to PMK). The funders had no role in study design, data collection and analysis, decision to publish, or preparation of the manuscript.

**Competing interests:** The authors have declared that no competing interests exist.

inflammatory profiles varied at admission and partially reverted to healthy range at EoT. Partial least squares determination and logistic regression showed that concentrations of inflammatory markers, including soluble TNF receptors and sCD40L, consistently changed between admission and EoT in all four countries, and were associated with increased odds of hepatomegaly and splenomegaly.

## Conclusions/significance

The recovery of haematological parameters, alongside a reduction in systemic inflammatory markers may be indicative of successful treatment of VL in East Africa. The biomarker dynamics suggest a partial resolution of inflammation and restoration of immune homeostasis during treatment. To confirm their predictive value, these markers should be evaluated in cohorts with a larger number of patients who experience treatment failure.

### Author summary

Visceral leishmaniasis (VL) is a life-threatening disease caused by infection with *Leishmania* parasites. It mainly affects vulnerable groups such as malnourished children, young adults, and people with HIV. It is endemic to South America, Asia and Africa, and most cases now occur in East Africa. While some people recover, others relapse or die, especially if they have weakened immune systems. Although treatment exists, little is known about how inflammatory markers change during and shortly after therapy. ImmStat@cure studied patients from Ethiopia, Kenya, Sudan, and Uganda to understand how their health and immune responses change from hospital admission to the end of treatment. By analysing blood samples and clinical data, we found that while some signs of illness improved by the end of treatment, others did not fully return to normal range. Variations in certain blood markers linked to inflammation, such as soluble TNF receptors and sCD40L, changed after treatment and were linked to symptoms like enlarged liver and spleen. These findings suggest that tracking these markers may help doctors understand how well a patient is recovering. Future research should test whether these indicators can predict which patients are at risk of not fully recovering or relapsing.

## 1 Introduction

Visceral Leishmaniasis (VL), also known as Kala-azar, is the most severe form of leishmaniasis [1,2]. It is mainly caused by parasites from the *Leishmania* donovani complex, *L. donovani* and *L. infantum*, and transmitted by female sand fly vectors. VL is endemic to more than 60 countries, and around 90% of the reported cases occur in Brazil, Ethiopia, India, Kenya, Somalia, South Sudan, and Sudan [2,3]. Following the

success of the tri-national VL control initiative in India, Nepal and Bangladesh [4–6], East Africa (EA) has emerged as the primary global hotspot of VL. As of 2022, EA accounted for 73% of the global human VL burden, with an estimated annual incidence of 8,000–13,000 cases and average case fatality rate of 4–11% [7,8]. Given this significant epidemiological burden, there is a pressing need to prioritize research and control efforts in East Africa.

Most *L. donovani* infected individuals are asymptomatic carriers [2,9,10]. When symptomatic, VL is characterized by splenomegaly, hepatomegaly, weight loss, pancytopenia, hypergammaglobulinemia, irregular fever, high parasitaemia and parasite burden, with consequent chronic inflammation in the liver, lymph nodes, spleen and bone marrow. VL has a mortality of ~95% in symptomatic patients who do not receive treatment [2,11]. There is substantial variation among symptomatic cases regarding disease progression, response to treatment, relapse and occurrence of post-kala-azar dermal leishmaniasis (PKDL) [12–14]. Host and parasite genetics are likely important factors for disease severity [12–14], including polymorphisms in the HLA locus [13,15,16]. In EA, the highest incidence of symptomatic VL is observed among malnourished children and young adults, and HIV co-infected patients, as a consequence of both immunological naivety/ suppression and behaviours that increase exposure to *Phlebotomus* vectors in endemic settings [17–20]. There is also evidence of gender disparity in VL cases, which might relate to behaviour and/or biological differences between sexes [21–23]. However, the underlying causes for this heterogeneity are not fully understood.

VL relapses are often associated with HIV co-infection. In Ethiopia, more than 50% of the VL-HIV coinfected patients relapse within 9 months of the end of the treatment [24], maintaining high parasite loads, hepatosplenomegaly and pancytopenia at the end of treatment cycle [25]. VL-HIV relapses were associated with failure to restore antigen-specific production of IFNγ, persistent lower CD4+T cell counts and higher expression of PD1 on CD4+ and CD8+T cells [25]. In HIV negative VL patients in EA, relapse rates were estimated as 5–13% depending on the treatment regime, and often occur within 6 months after the end of the treatment [26–29]. In South Sudan, spleen size on admission and persistent splenomegaly at the end of treatment with sodium stibogluconate and paromomycin (SSG/PM) were associated with a higher risk of relapse, while no associations were observed with age, sex, malnutrition and treatment complications [30]. Similarly, spleen size at discharge was associated with relapse risk in patients treated with liposomal amphotericin B [31], but not with Miltefosine in India [32]. Evidence of parasite persistence or poor anti-leishmania immune responses after the end of treatment were also associated with treatment failure in VL patients. Higher concentrations of IgG1 six months after the end of treatment were associated with a higher relapse in Indian patients [33], while parasite load after 56 days of the start of the treatment, measured by qPCR, may be a sensitive VL relapse marker in EA [34].

In rodent models of VL drug treatment, splenomegaly and hepatomegaly are still evident at the end of treatment, and transcriptomic analyses have shown that indicators of systemic abnormalities of immune, metabolic and tissue remodelling processes were not fully restored to homeostasis [35–37]. Most clinical studies evaluating patients at admission and end of treatment (EoT) in EA have been restricted to one country and used limited marker panels. In Ethiopia, IL-10 and IFNγ were identified as potential markers for active VL, with IL-10 concentration declining within seven days of treatment [38]. In Kenya, severe VL was also associated with higher circulating IL-10 and IFNγ, with reductions in both at 17 days post treatment with SSG/PM, despite persistent anaemia and circulating IL-17 [39]. In Sudan, a longitudinal study identified IFNγ and TNF concentrations as important markers for active VL. [40]. Hence, the broader systemic immune/ inflammatory status of patients across EA countries at the end of treatment with SSG/PM is not fully understood.

To address these shortcomings and further understand the links between hepatosplenomegaly and systemic inflammatory markers, we conducted ImmStat@cure, the most comprehensive multicentre clinical study in EA to investigate the clinical and immune response status in VL patients on admission and at EoT, after treatment with SSG/PM. The ImmStat@ cure consortium encompassed four EA countries (Ethiopia, Kenya, Uganda and Sudan). Though comprised of genetically diverse host populations [41,42], these countries are the focus for most drug trials conducted in EA, and show high prevalence for VL, and in the case of Sudan, PKDL. ImmStat@cure aimed to characterize patient immune response by several approaches, including serologic immune markers, T and B cell immunophenotyping, transcriptomics and proteomics.

Here, we first describe the study design and clinical characteristics of the patient cohort comprising ImmStat@cure. Using a panel of serologic inflammatory markers that have proven useful to understand the pathogenesis of other infectious and non-infectious systemic diseases [43–51], we identify both common and country-specific changes in inflammatory profile at EoT. We provide preliminary evidence to support sCD40L and sTNR receptors as prognostic markers of, and contributors to, the pathophysiology of hepatomegaly and splenomegaly.

## 2 Materials and methods

### 2.1 Ethical and regulatory approval

This multicentre clinical study was approved by ethical and regulatory agencies in the UK (Department of Biology Ethics Committee reference PK202001, Hull York Medical School Ethics Committee reference 22-23.58), Ethiopia (University of Gondar Ethics Committee reference V/P/RCS/05/495/2019; Ministry of Education reference 7/2–324/M259/35), Kenya (National Commission for Science, Technology and Innovation references NACOSTI/P/21/11096/P6958/25/415974; KEMRI Scientific and Ethics Review Unit reference KEMRI/SERU/CCR/0169/4113), Sudan (University of Khartoum Ethics Committee reference FM/DO/EC; Sudan Ministry of Health approval 7-12-20) and Uganda (Makerere University Ethics Committee reference SBS-REC-751;Uganda National Council for Science and Technology reference HS1266ES). All studies were conducted in accord with the Declaration of Helsinki and with participant informed consent/ assent. Formal written consent to participate in the study was obtained from the patients or their legal guardians. The study did not involve modification of existing standards of care provided to patients.

### 2.2 Study sites and recruitment

Immstat@cure was an observational multicentre clinical study that sought to recruit up to 160 VL patients to assess the clinical and immunological status before and at EoT after treatment with SSG/PM/MG across four countries in EA (40 per site). Up to 120 healthy convenience controls (healthy volunteers - HV; 30 per site) were also collected to provide indicative baseline data. Patients were recruited by active case detection and self-attendance at study hospitals. The study was prospectively registered on ClinicalTrials.gov (ClinicalTrials.gov ID: NCT04342715, link: https://clinicaltrials.gov/study/NCT04342715?cond=NCT04342715&rank=1). The protocol is provided in S1 Supplementary Information.

In Ethiopia, patient recruitment and study visits as well as all haematological and biochemical screening were conducted at University of Gondar (UoG). Healthy endemic controls were recruited from the Gondar region. In Kenya, VL patient recruitment and study visits were conducted at Chemolingot Sub-County Hospital in Baringo County, and healthy non-endemic controls were recruited at the Center for Clinical Research, Nairobi County, Kenya. Immunological assays were performed at the Kenya Medical Research Institute (KEMRI). In Sudan, patient recruitment, study visits, haematological and biochemical screening and safety tests were conducted at Soba Hospital, Khartoum, the Institute for Endemic Diseases (IEND), the El Hassan Centre for Tropical Medicine, Doka, Gedarif State and the University of Khartoum, Sudan. Healthy endemic controls were recruited from the same region. In Uganda, patient recruitment, study visits, haematological and biochemical screening were conducted at Amudat Hospital, and at the Makerere University (MU) College of Health Sciences. Healthy controls were recruited from the Kampala region.

VL diagnosis was initially performed using the rK39 rapid diagnostic test [52]. Confirmatory diagnostics was performed by a parasitologist through microscopic examination of aspirates (spleen, bone marrow or lymph nodes) stained with Giemsa and observed at 10x eyepiece and x100 magnification, to identify amastigotes. Aspirates were graded from 1+ to 6+, based on the WHO TRS 949 guidelines [53]. Inclusion criteria are fully described in the Protocol (S1 Supplementary Information) and included: Aged 12–50 years; either sex; confirmed VL diagnostics; suitable for treatment using SSG/PM; negative for malaria, tuberculosis, leprosy, HIV, HBV, HCV; no previous diagnosis of leishmaniasis; not pregnant or lactating; and not severely malnourished, based on country-specific metrics. Patients were treated with SSG (20mg/Kg/day;

i.m.) and PM (15 mg/kg/day; i.m.) for 17 days. The occurrence of relapse and PKDL was assessed at 6 months follow up after completing treatment.

## 2.3 Sample and data collection

Clinical, haematological and immunological data were collected on admission and at the end of treatment. Demographic data, sex, age, height, weight; and clinical data: sitting systolic blood pressure (SSBP), sitting diastolic blood pressure (SDBP), temperature, pulse; skin, cardiovascular or respiratory abnormalities; the presence of hepatomegaly, spleno-megaly and lymphadenopathy were collected by clinicians. Blood was also collected on both time points, for biochemical and immunological assays. Details of standard reference ranges for blood biochemistry are included in the Protocol (S1 Supplementary Information). Blood samples for bulk transcriptomics and proteomics, as well as for CD8+T-cell and B-cell phenotyping by flow cytometry were also collected. These will be evaluated and reported in future publications.

## 2.4 Plasma inflammation markers

Inflammatory markers were quantified by flow cytometry, using the LEGENDplex Inflammation Panel 2 (BioLegend) kit on a Beckman Coulter Cytoflex flow cytometer, following manufacturer instructions, using plasma obtained from peripheral blood. The evaluated markers were: CX3CL1, CXCL12, PTX3, TGF.B1, sCD25, sCD40L, sRAGE, sST2, sTNF.RI, sTNF. RII and sTREM.1. The LEGENDplex Data Analysis Software Suite (Qognit, v2024-06–15, San Carlos, CA, USA) was used for initial data processing and transformation from median fluorescence intensity (MFI) to predicted concentration. For samples from each country, the highest value of the Limit of Quantification (LOQ) across experimental plates for each marker was used as a lower cutoff value for the measurements across all plates. Measurements that were below this cut-off were imputed as this lower LOQ cutoff. Samples were analysed in duplicate, and the mean of the duplicates was used for downstream analysis. To account for inter-plate variation, pooled control VL patient samples were included in each plate from the same country. Marker values were standardized by scaling each measurement to the mean value of the corresponding markers across VL control pools. Statistical analysis was done in R (see below).

## 2.5 Identification of markers for hepatomegaly and splenomegaly

Potential markers for hepatomegaly and splenomegaly were identified using a combination of logistic regression and par-tial least squares discriminant analysis (PLS-DA). Four binary clinical outcome groups were evaluated: i) Hepatomegaly at admission (V1); ii) Persistent splenomegaly after treatment (V2); iii) Splenomegaly at admission (V1); and iv) the accu-racy of pre-treatment clinical/immunological markers (V1) to predict persistent splenomegaly after the end of treatment (V2). Due to data heterogeneity, each country was analysed independently with comparisons as reported in the Results. Patient data was log-transformed log10(x+0.001) and scaled, based only on the dataset used. Only patients where all the assessed clinical, haematological and inflammatory markers were present were evaluated.

Markers that increased the odds of the clinical outcomes were identified by logistic regression, optimized to unbalanced and low sample sizes, with the *logistf* function in R [54], using the family "binomial" (as in presence or absence of hepato-megaly) and the method "logistf" (Firth's bias-reducing penalized likelihood to reduce small-sample bias). Representative markers were selected based on p-value<0.05 and a confidence interval for the regression coefficient that did not cross zero, as this would indicate a reversal of the direction of the association (positive vs. negative odds). Prediction uncer-tainty was assessed using 1000 stratified bootstrap replicates, maintaining the original case-control ratio by sampling the same number of cases and controls as in the original dataset. Logistic regression evaluations were also performed using age as a covariate.

In parallel, the predictive power of each marker for the clinical outcomes was assessed using partial least squares dis-criminant analysis (PLS-DA), implemented via the mixOmics and caret packages in R [55]. Initially, Variable Importance in Projection (VIP) scores were calculated to rank markers according to their contribution to discrimination of the outcomes.

Subsequently, models incorporating the top 2, 3, 4, 5, 6, 10, or 15 markers (based on VIP scores) were evaluated using PLS-DA with leave-one-patient-out (LOPO) cross-validation to assess their predictive performance. The pre-selection of markers might lead to data leakage and an overestimate of the model predictive value, but ensures that each LOPO iteration uses the same markers. Concordance between the markers identified by logistic regression and those highlighted by PLS-DA increases confidence in their relevance to the clinical outcomes under investigation.

### 2.6 Statistical analysis and data representation

All comparative statistical analysis and data representation were generated in R [56]. Differences between VL patients pre and post treatment, and for the clinical, haematological immunological markers were assessed using non-parametric paired Wilcoxon signed rank test, while comparisons between VL patients and convenience controls were performed using the Wilcoxon rank sum test (equivalent to Mann-Whitney test). P-values were adjusted for multiple comparisons using Benjamini-Hochberg (BH) false discovery rate (FDR) method. All patients that had data for a given marker were included in the evaluation, even if there were missing data for other analytes. The number of patients used in each comparison and the min-max values for each clinical trait from each country can be seen in S1 and S2 Tables. For radar plots, we first estimated the median value for each marker in each condition (pre-treatment V1, post-treatment V2 and convenience control HV). For each marker, the highest median value for the three conditions was set to 100%, and the remaining values were scaled proportionally. The Spearman correlation between and within clinical, haematological and immunological markers was estimated in R using cor.test, and the FDR was corrected using BH.

For the principal component analysis (PCA) and Uniform Manifold Approximation and Projection (UMAP), the data was log-transformed $\log10(x+0.001)$, centred (subtracting the mean) and scaled (divided by the standard deviation). Only samples with the complete set of markers were used. Then, the prcomp and umap [57] functions were respectively used to generate the PCA and UMAP. The UMAPs were generated using the following parameters: min_dist=0.1, metric="euclidean", n_epochs=1000, n_neighbour=15, seed=123. Graphical representations were generated using ggplot and factoextra. For the country specific Inflammation data PCA/UMAPs, the used markers were: CX3CL1, CXCL12, PTX3, TGF.B1, sCD25, sCD40L, sRAGE, sST2, sTNF.RI, sTNF.RII and sTREM.1.

## 3 Results

### 3.1 Study overview and patient demographics

ImmStat@cure was a observational multicentre clinical two time-point study to assess the immune response status in VL patients on admission before treatment (V1), and ~17 days after treatment (V2; EoT). Clinical, haematological and inflammatory markers data were collected from patients on both time points and from convenience controls at a single time point. A total of 212 patients were enrolled, corresponding to 109 VL patients, and 103 convenience controls (HV) across the four countries. Patients were recruited during the course of 3 years, where the first and last recruitment for each country were: Ethiopia, 01/2024 and 06/2024; Kenya, 11/2022 and 04/2024; Sudan, 01/2023 and 05/2024; Uganda, 02/2022 and 06/2024. Recruitment is shown in the flow diagram (Fig 1).

The occurrence of relapses or PKDL in VL patients was assessed 6 months after the end of the treatment. Only 3 relapses were reported. The patient retention was ~98%, with variable concentrations of data collection completeness (Figs 1 and S1 Fig and S1 Table). Patients in Ethiopia, Kenya, Uganda and Sudan had a median age of respectively 21.5, 15, 20 and 16 years, and respectively 100%, 88%, 72% and 77% were male. All patients were serologically negative for malaria, HIV, Hepatitis B, Hepatitis C, HBV, HCV. All VL patients were treated for leishmaniasis as per standard of care using (SSG/PM). Five patients in Ethiopia switched drug treatment to Ambisome after clinical abnormalities related to SSG/PM were noted. One patient from Ethiopia died during treatment due to visceral leishmaniasis complications. Only patients with missing data were not included in each downstream comparison (see methods).

PLOS Neglected Tropical Diseases

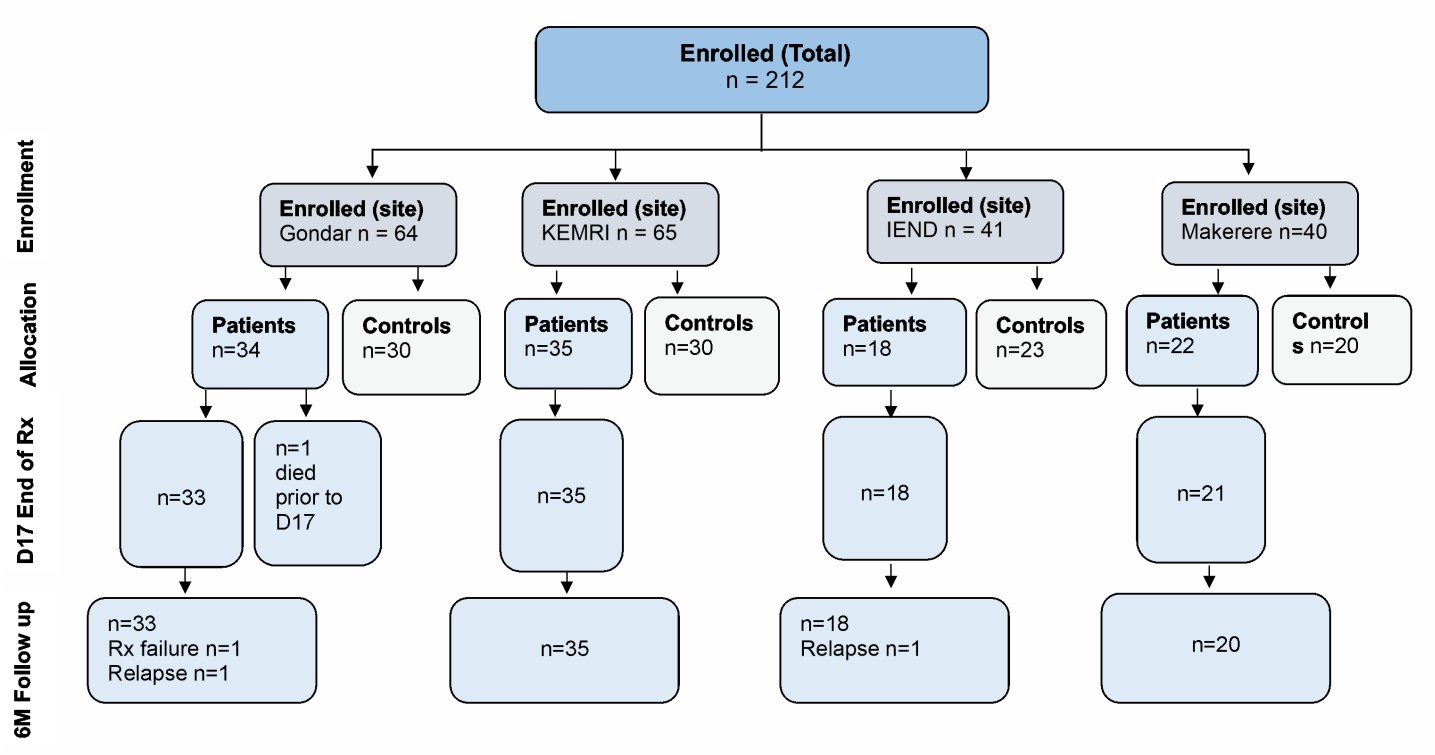

**Fig 1. Flow diagram summarizing the Immstat@cure study.** A summary of the patient demographics and clinical state can be seen in S1 Table. Ethiopia (University of Gondar), Kenya (Kenya Medical Research Institute - KEMRI), Sudan (Institute of Endemic Diseases) and Uganda (Makerere University).

### 3.2 Clinical and haematological markers before and at the end of VL treatment in East Africa

An evaluation of hepatomegaly, splenomegaly and 18 clinical and haematological traits across East Africa showed that VL patient clinical markers recovered towards normal reference range (NRR) by the end of treatment, with the extent of improvement varying across countries. Splenomegaly was observed in approximately 70–100% of patients at admission, but decreased to 0–5% in Sudan and Uganda, and to 31% and 88% respectively in Kenya and Ethiopia. Hepatomegaly was reported at admission in Ethiopia (26%), Sudan (38%) and Kenya (~3%), but by the end of treatment, only one case persisted in Ethiopia (Fig 2A). Similar improvements were observed for the majority of the haematological markers, with recoveries in albumin and haemoglobin concentrations; and platelets, white blood cell (WBC), neutrophils and lympho-cytes counts after treatment (Figs 2B and S2 Fig and S2 Table).

Ethiopia had the lowest average proportion of patients with markers within Normal Reference Range (NRR) on admission (~30%), when compared to Kenya (~54%), Sudan (~70%) and Uganda (~55%). This suggests that the patients were in different stages of VL when hospitalized, with more severe cases in Ethiopia. Ethiopia also had the largest recovery after treatment, with an average of ~70% of the patients within NRR, which was comparable to the ~76% in Kenya and ~84% in Sudan. Some of the largest recoveries to NRR were for albumin concentrations (~6% V1 to ~60% V2), platelets (~8% V1 to ~88% in V2) and WBC counts (~6% V1 to ~70% in V2) in Ethiopia; Albumin (~28% V1 to ~66% V2) and Hae-moglobin concentrations (~9% V1 to ~51% in V2) in Kenya and platelet counts (~39% to ~83%) in Sudan (Figs 2C and S3, S4 and S5 Figs and S3 Table). This likely reflects liver, spleen and bone marrow recovery and immune control after effective VL treatment in the four countries.

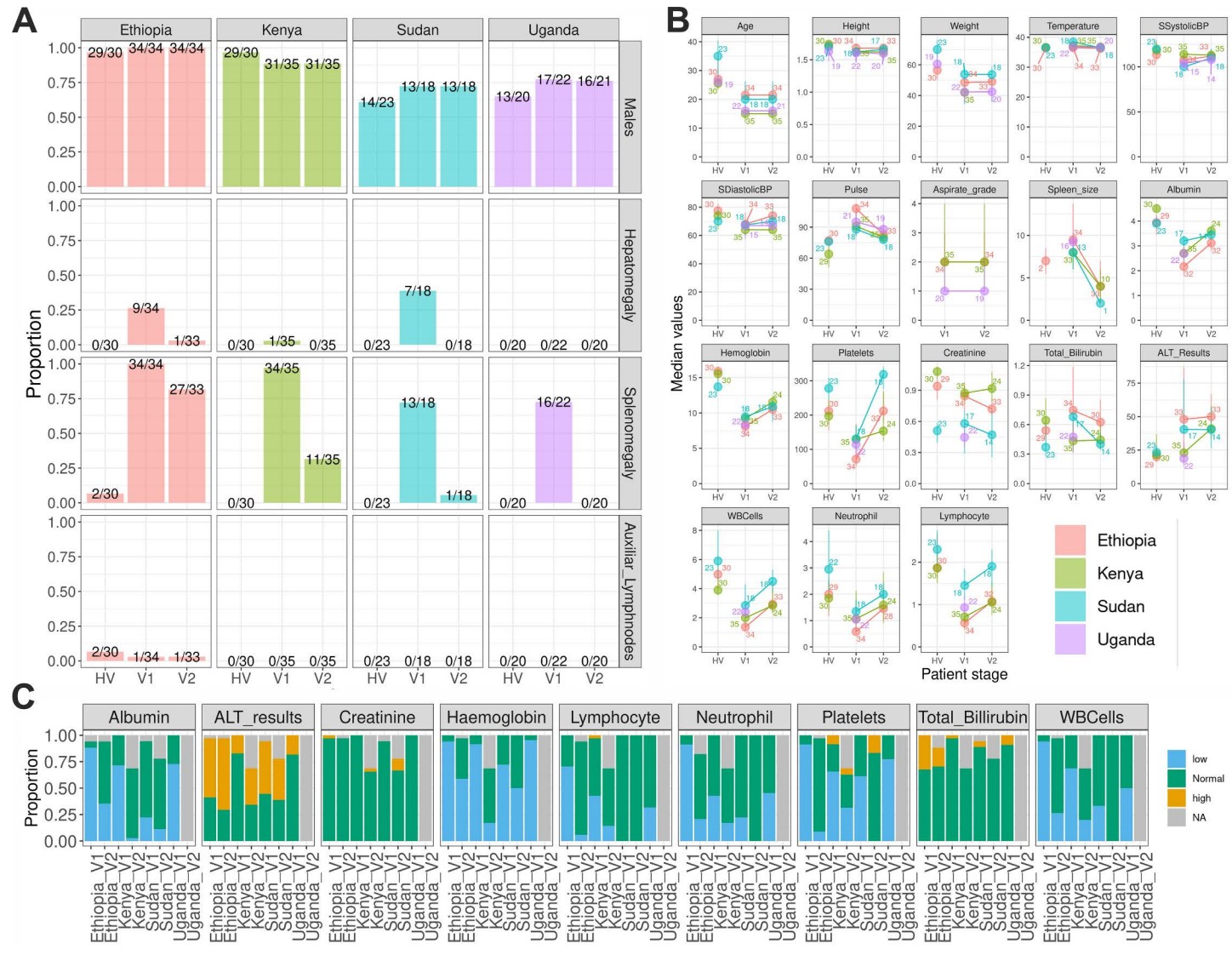

**Fig 2. Patient clinical symptoms and haematological markers before and after treatment for VL. A)** Proportion of the male patients, and patients with hepatomegaly, splenomegaly or detectable axillary lymph nodes. The columns represent convenience controls (HV), and patients pre-treatment (V1) and post-treatment (V2). At the top of each bar, the number on the right corresponds to the total of the samples that had non-missing data, while the number on the left corresponds to the number of samples that were positive for the clinical trait. **B)** dot and whiskers plot representing the median, 25 and 75 quantiles for each continuous clinical trait/haematological marker assessed in the project. Each box represents a different clinical trait/haematological marker, each colour represents a different country and the number corresponds to the number of evaluated samples. **C)** Proportion of patients' clinical traits that were below (blue) within range (green) or above the normal range (gold) for each haematological marker accessed. Missing data is represented in gray. Statistical comparisons between V1, V2 and HV results separated by sex can be seen in S2 Fig and S2 Table. Interval ranges for each patient group can be seen in the S1 Table.

The lack of post-treatment clinical and haematological data for Uganda prevented most of the V1-V2 comparisons in this country. Due to the limited sample size, the female dataset also lacked sufficient statistical power to detect significant differences before and after treatment, although recovery trends similar to the male dataset were observed (S2E-G Fig and S2 Table). The limited female sample size also reduced the statistical power for male–female comparisons, where only higher creatinine concentrations in males pre-treatment (V1) from Sudan reached statistical significance (S6 Fig).

### 3.3 Changes to inflammatory markers pre and post treatment

To evaluate the inflammation status of each patient before and after VL treatment, we used a panel of 11 inflammatory markers. This includes soluble decoy receptors for Tumour Necrosis Factor (TNF) (sTNF-RI and II) [46,58]; sRAGE, the soluble decoy for Receptor for Advanced Glycation End-products (RAGE) [47,59,60]; potential markers for T-cell activation (sCD25 - IL-2Ra) [48,61]; myeloid cell activation and septic shock (soluble Triggering Receptor Expressed on Myeloid Cells-1 - sTREM-1) [62,63]; cell trafficking (CXCL12 - SDF1); [64] acute inflammation (Pentraxin 3 - PTX3) [65]; and chronic inflammation with macrophage and B-cell activation (sCD40L) [66,67]. Most of the analytes were above the limit of detection (S2 Fig), with the exception of TGF β1 in all countries, and sTREM1 and CX3CL1 in Kenya. The standard curves were similar between plates, and the Limit of Quantification (LOQ) used for each country can be seen on S7 Fig.

The combined analysis of 11 inflammatory markers suggests that post-treatment patients are in an intermediate state between pre-treatment and convenience controls for the four countries (Fig 3 and S8 Fig), consistent with the clinical traits. For male patients, Principal components 1 and 2 collectively accounted for around 50% of the overall data variability, indicating that they effectively captured the main patterns in the data (S8 Fig). Comparable tendencies were observed for female patients (S9 Fig), although with lower statistical support.

Plasma concentration of the inflammatory markers sTNF-RI, sTNF-RII, sST2, and sCD25 were consistently higher prior to treatment, suggesting robust inflammatory activity characterized by TNF production, myeloid and T cell activation during active VL. These concentrations declined following treatment, concomitantly with an increase in sCD40L and, to a lower extent sRAGE, concentration. Taken together, this suggests a decrease in the immune activation status after treatment, but with persistent chronic inflammation (Figs 3, S10 Fig and S11 Fig and S2 Table).

Comparisons between males and females were again limited by the small female sample size. However, PTX3 concentration was significantly higher in males after treatment in Sudan, whereas sRAGE concentration was higher in females following treatment in Uganda (S6 Fig),

When each stage, pre- or post-treatment, was evaluated independently, few significant correlations after FDR correction were observed between clinical traits, haematological and inflammatory markers, aside the expected weight-height and white blood cells - neutrophils associations (Figs 5 and S12 Fig and S4 Table). sTREM-1/CX3CL1 and sTNF.RI/sCD25 were correlated in Ethiopia and Uganda pre-treatment, and in Kenya and Uganda post-treatment. sTNF.RI/CX3CL1 was correlated in Ethiopia and Uganda pre-treatment, while sTNF.RI/sTNF.RII was correlated in Kenya and Uganda post- treatment.

Within the cohort, two patients from Ethiopia, VL006 and VL016, and one patient from Sudan, VL017, relapsed (Fig 4). From those, VL006 relapsed just after treatment, while VL017 relapsed within 6 months and VL016 relapsed 6 months after the treatment. The small number of relapsed cases in each site limits the statistical power to identify associations between clinical/immunological markers and relapse. However, some indicative patterns were observed. VL006 and VL016 were the only Ethiopian patients with a grade 6 + spleen aspirate at admission (Fig 4A), supporting a potential association between the parasite burden before treatment and increased risk of relapse. There was no aspirate grade data for Sudan samples. Patient VL006 did not exhibit reductions in the concentrations of the inflammation markers sTNF-RI, sTNF-RII, and sCD25, neither a large recovery in albumin concentration following treatment, in contrast to the majority of cured visceral leishmaniasis (VL) patients in Ethiopia. In fact, VL006 had the highest values of both sCD25 and sTNF.RII at the EoT. Patient VL017 from Sudan also had low albumin concentration, the highest level of sCD25 and one of the highest concentrations of sTNF-RI at EoT. Similar results were not observed for VL016, suggesting that there were differences in the immune state of relapsed patients. The difference in immune response is also supported by the fact that hepatomegaly at admission was only observed in VL006. Both Ethiopian patients exhibited some of the lowest values for sRAGE and the highest values for CX3CL1 and sTREM1 on both pre and post treatment time points, suggesting that these patients still have persistent inflammation, with signs of myeloid activation and recruitment to inflamed tissues at the EoT.

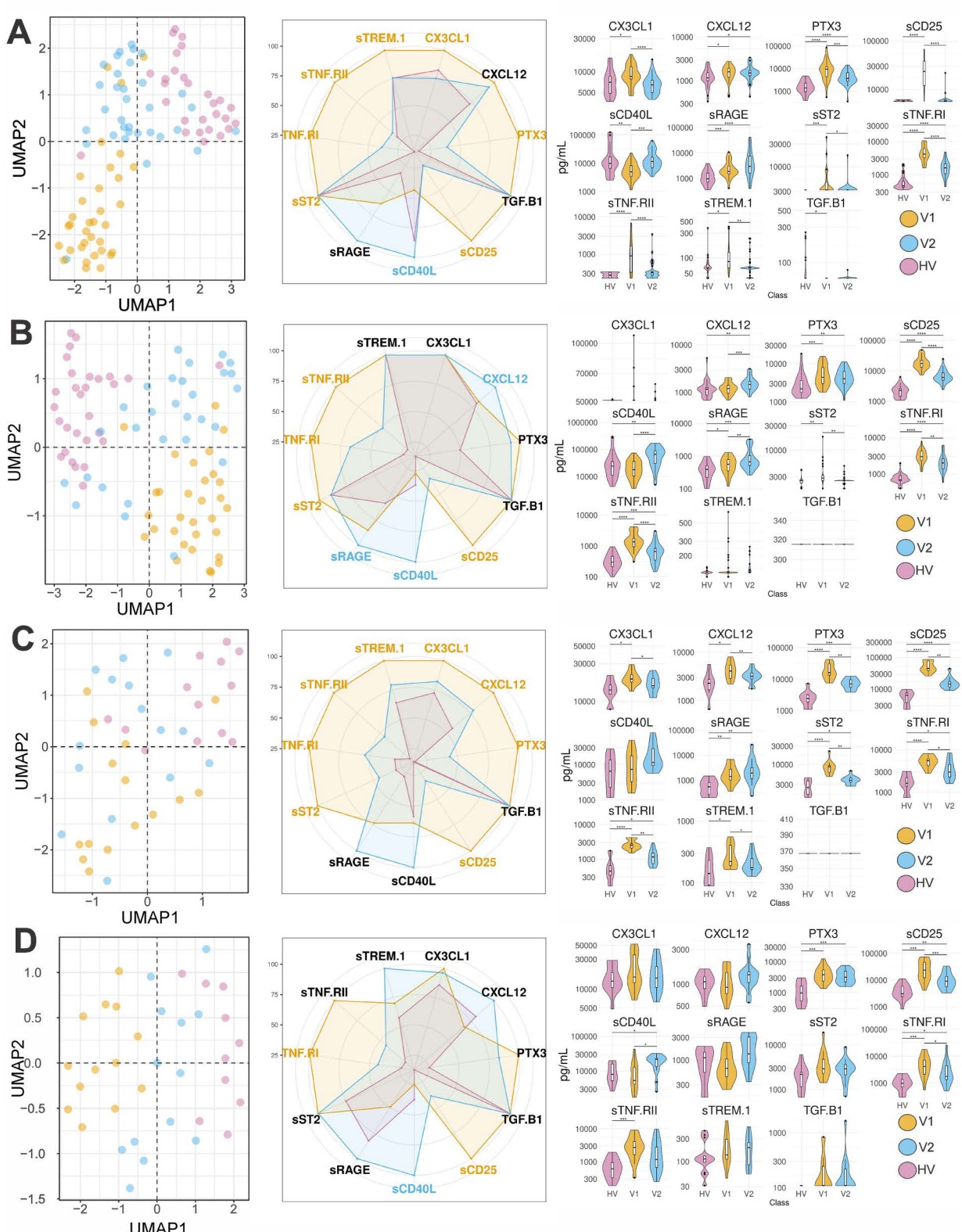

**Fig 3. Inflammatory markers in VL male patients across East Africa. A-D**, Inflammatory markers for patients from Ethiopia **A)**, Kenya **B)**, Sudan **(C)** and Uganda **(D)**. **Left panel:** UMAP of the inflammatory marker values for each patient. Each patient is represented by a dot, coloured in yellow (V1)

blue (V2) and red (HV). The UMAP was generated with samples that had the complete set of inflammatory markers (Ethiopia: HV:29, V1:34, V2:34; Kenya HV:29, V1:31, V2:31; Sudan HV:14, V1:13, V2:13; Uganda HV:14, V1:13, V2:13). PCAs with the same data can be seen in S8 Fig. **Middle panel:** radar plot, where each segment corresponds to the median of patients in a group (V1, V2 or HV), scaled to the highest median in the three groups of one inflammatory marker. Markers that were significantly (p < 0.05) higher in V1 or V2 are respectively coloured in yellow or blue. Markers that were not significantly different are in black. **Right panel**: Violin plots showing the range and statistical support for the variations in inflammatory markers pre and post treatment. The numbers of asterisks represent p-values, where 1 to 4 corresponds respectively values below 0,05, 0.01, 0.001 and 0.0001. The statistical results and number of samples used in each comparison can be seen in S2 Table.

### 3.4 Prognostic markers of hepatomegaly and splenomegaly

To identify prognostic markers of hepatomegaly and splenomegaly and their change associated with treatment, we used logistic regression and partial least squares discriminant analysis (PLS-DA). Potential markers associated with pre-treatment hepatomegaly and early markers for persistent splenomegaly in Ethiopia and with persistent splenomegaly in Kenya were identified (Fig 5 and S5 Table). Given the small and unbalanced sample sizes, these markers should be interpreted with caution, rather than definitive predictors of these outcomes.

For Ethiopia, from the 31 patients with complete data, 9 had hepatomegaly. Logistic regression analysis identified sTNF.RII and albumin concentrations respectively associated with higher and lower odds of having hepatomegaly. sTNF.RII association was supported even when using age as a covariate (S13 Fig). PLS-DA analysis identified five markers with VIP values above 1.3, albumin, sTNF.RII, sCD40L, hemoglobin and ALT concentrations, in accordance with the logistic regression results. LOPO validation of the predictive model using the top 3 VIPs, albumin, sTNF.RII and sCD40L resulted in the highest balanced accuracy of 79% and Negative Predictive Value (NPV) of 53% (Figs 5A and S14A, and S5 table and Man Whitney U test p-values on S6 Tables).

Next, to evaluate the value of pre-treatment data to predict the persistent splenomegaly (splenomegaly after treatment), a total of 31 patients, 27 with persistent splenomegaly from Ethiopia, were used. The logistic regression analysis identified lower sCD40L concentration as associated with higher odds of developing persistent splenomegaly. sCD40L association was significant even when using age as a covariate (S13 Fig). The PLS-DA analysis identified four markers with VIP higher than 1.3, sCD40L, CXCL12, sST2 and platelet counts. LOPO validation of a model with the top two VIPs, sCD40L and CXCL12, resulted in the highest balanced accuracy of 94%, and a PPV of 83% (Figs 5B and S14 Fig and S5 table and Mann-Whitney U test p-values on S6 Tables).

Finally, for persistent splenomegaly in Kenya, from the 29 patients with complete data, 10 had splenomegaly after treatment. The logistic regression analysis identified albumin concentration, WBC and neutrophil counts as associated with lower odds and sTNF.RI as associated with higher odds of developing persistent splenomegaly. Albumin and sTNF.RII associations were still significant even when using age as a covariate (S13 Fig). PLS-DA analysis identified five markers with VIP higher than 1.3: albumin, WBC, sTNF.RI, neutrophils and sCD25. LOPO validation of a model with the top 4 VIPs, albumin, WBC, sTNF.RI and neutrophil counts resulted in the highest accuracy of 89% and a PPV of 86% (Figs 5C, S14 Fig and S5 table Mann-Whitney U test p-values on S6).

Results for hepatomegaly in Sudan, early splenomegaly in Uganda and Sudan, persistent splenomegaly in Ethiopia and the predictive power of early markers to identify late splenomegaly in Kenya can be seen in S15 Fig and S5 Table. For the Sudan and Uganda sets, the sample number was low, which might result in lower precision and inflated PLS-DA predictive power and logistic regression odds ratios.

Taken together, these results suggest that higher concentrations of sCD40L and a reduction in the concentrations of sTNF receptors might be important for a better prognosis of VL after treatment, with reduced odds for hepatomegaly and persistent splenomegaly.

### Discussion

Immstat@cure is a multi-institute project that aims to characterize the clinical and immunological status of VL patients at admission and at the EoT, across EA. In this initial phase of analysis, by integrating detailed clinical data with serological

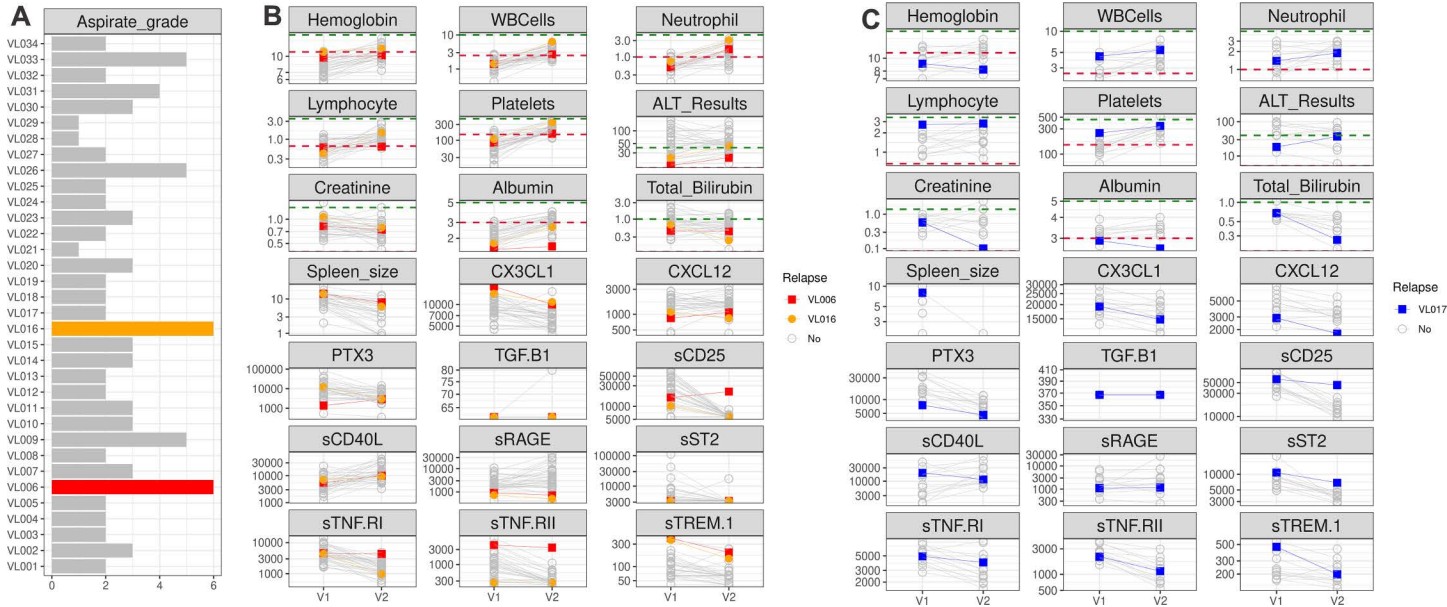

**Fig 4. Clinical and immunological features of relapsed patients.** Only three patients relapsed. Two from Ethiopia, VL006 and VL016, and one from Sudan VL017. **A)** Spleen aspirate grade of all patients from Ethiopia. **B)** and **C)** Slopegraph showing values for clinical and inflammatory traits before (V1) and after (V2) treatment among patients from Ethiopia. VL006, VL016 and VL017 are represented respectively in red, orange and blue, while other VL patients from the same country are represented by gray circles. For the clinical traits, normal reference ranges are represented by green (upper normal limit) and red (lower normal limit) horizontal dashed lines.

and inflammation markers from VL patients across four East African countries, we have shown that: (1) patient clinical and inflammatory profiles varied at admission and partially revert to healthy values at EoT, with different degrees of recovery across countries; (2) concentrations of inflammatory markers, including soluble TNF receptors and sCD40L, consistently changed between admission and EoT in all four countries; and (3) these markers are also associated with increased odds of hepatomegaly and splenomegaly in visceral leishmaniasis (VL). This is the most comprehensive multi-country assessment of VL patient clinical and inflammatory state in EA to date.

At admission, splenomegaly, hypoalbuminemia, anaemia, and pancytopenia were widely observed in VL patients from the four countries, while hepatomegaly was observed in cases from Ethiopia and Sudan. In VL, albumin concentrations are typically low due to chronic inflammation and liver dysfunction, while decreased haemoglobin and white blood cell counts can be caused by systemic and splenic inflammation [68] and bone marrow dysfunction [69]. This observation is consistent with the elevated concentrations of inflammatory markers, such as sTNF-RI, sTNF-RII, sCD25, and sST2, detected in EA VL patients at admission. These markers collectively indicate a systemic immune/ inflammatory response characterized by a pronounced pro-inflammatory profile, including activation of T cells, with suppression of the Th2 response and a concomitant shift toward a Th1-dominant immune profile at patient admission.

After 17 days of treatment, substantial improvements were observed in the patients' clinical condition, with splenomegaly only persisting in Ethiopia and to a lower extent in Kenya, while hepatomegaly was no longer evident except for one patient in Ethiopia. This was concomitant with a decline in sTNFR, sCD25 and sST2 concentrations; and an increase in sCD40L and sRAGE concentrations, suggesting a decrease in the immune activation status at EoT, but with persistent chronic inflammation. This partial homeostasis restoration is in agreement with what was observed in a transcriptomics analysis of spleen and liver from *L. donovani* infected mice before and after treatment with AmBisome [37]; in microarrays from blood from VL patients from India treated with 15 lower doses of amphotericin B over 30 days, or a single high dose

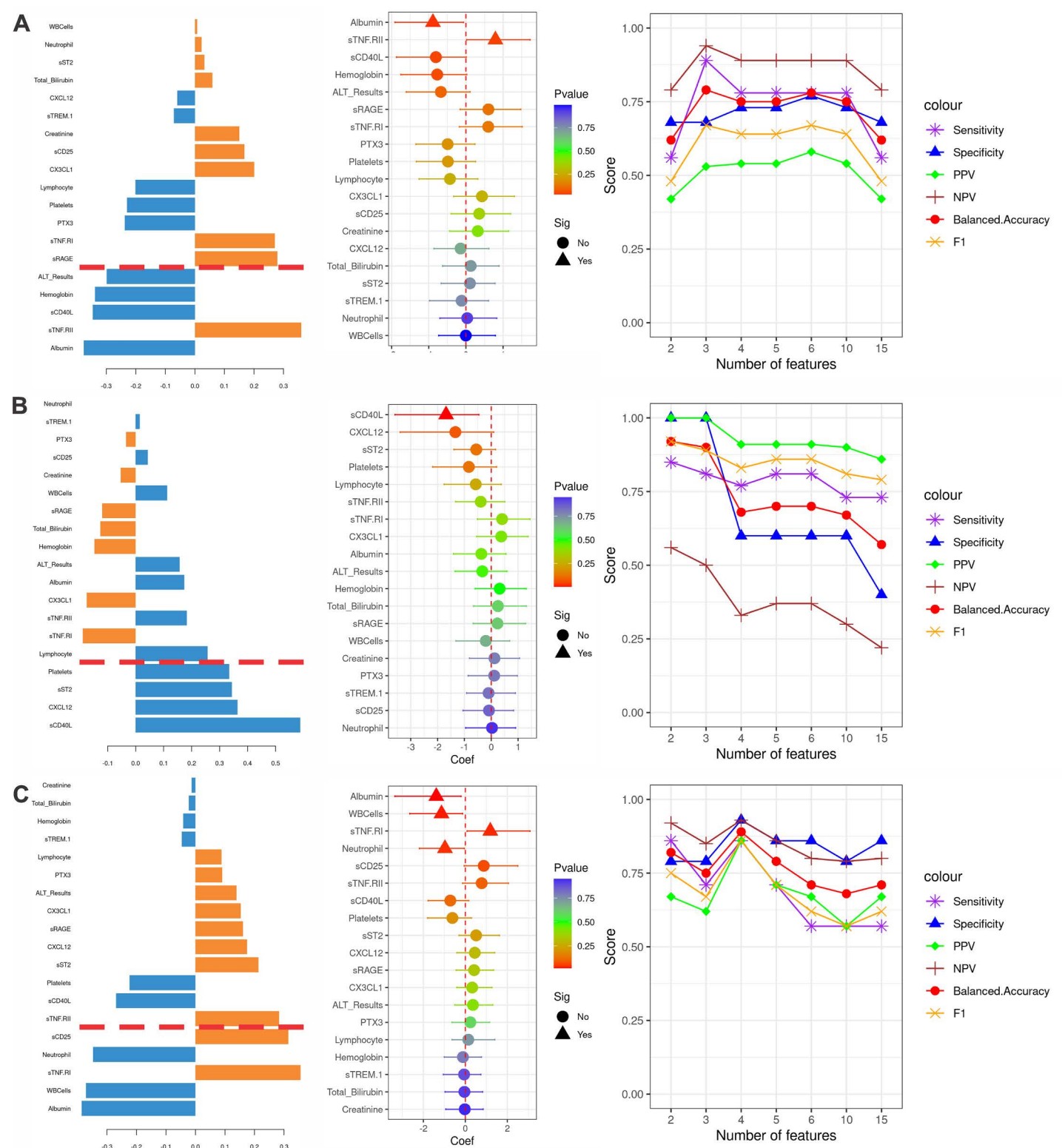

**Fig 5. Hepatomegaly and Splenomegaly (traits) marker assessment. A-C,** Trait assessment for Ethiopia hepatomegaly in pre-treatment patients
**(A)**, Ethiopia, prediction of persistent splenomegaly based on pre-treatment data **(B)** and Kenya persistent splenomegaly post-treatment **(C)**. **Left panel**:

sPLS-DA loading plot, showing the loading weight of the Inflammatory/Clinical markers influence on the assessed trait, ordered from bottom (highest impact) to top (lowest impact). Values in orange and blue are respectively associated with the presence or absence of the trait. Inflammatory/Clinical markers below the red line had Variable Importance in Projection (VIP) Values higher than 1.3. **Middle panel**: Logistic Regression coefficient with 95% confidence intervals for the association of each marker to the evaluated trait. Increases in values of markers that are above or below zero respectively increases or decreases the Odds Ratio of the trait. Values with p-value <0.05 are presented by triangles. Bootstrap validation of these predictions can be seen in the S14 Fig. **Right panel**: Model prediction scores result for the PLS-DA leave one patient out (LOPO) using increasing numbers of markers, selected based on highest VIP scores. PPV: Positive Predictive Value; NPV: Negative Predictive Value.

of liposomal amphotericin B at admission [70]; and in microarrays from VL patients from Brazil, treated with pentavalent antimony [71]. Taken together, these results indicate that, regardless of the treatment administered for visceral leishmaniasis (VL), patients often do not achieve full homeostatic restoration at EoT.

From the eleven evaluated inflammation markers, six have potential to be used to assess treatment efficacy: high concentrations of sTNF-RI, sTNF-RII, sCD25 and sST2 at admission and high concentrations of sCD40L and sRAGE at EoT. sTNF receptors were identified as potential markers for treatment success in VL across EA and as potential markers for hepatomegaly in Ethiopia and persistent splenomegaly in Kenya. These findings are consistent with reports of sTNF receptor concentration in VL patients treated for 30 days with SSG in Sudan [72]; or meglumine antimoniate in Brazil [58]. A reduction in serum concentrations of sTNF receptors is also a marker for treatment success in tuberculosis infections [73]. High concentrations of sTNF receptors were associated with severity, acute kidney injury and mortality in patients with COVID-19 [74], and with active infection and liver damage in schistosomiasis japonica [75]. Similarly, a reduction in concentrations of sCD25 and sST2 could also indicate VL treatment success in EA. Reduction in sCD25 concentrations was observed after treatment in VL patients from Bangladesh [76], as well as in patients with brucellosis [77]. sST2 is increased in patients with pathogenic inflammation [78–80] and sepsis [81–83]. Elevated sCD40L concentrations were observed in VL patients after antimony treatment, with concentration inversely correlating with spleen size and parasite burden [51]. Moreover, *in vitro* studies have demonstrated that recombinant CD40L, as well as human sera with high concentrations of sCD40L, reduced both the number of infected macrophages and the intracellular load of *Leishmania* amastigotes [67]. Similarly, low sCD40L concentration has been observed in HIV–VL coinfected patients compared to HIV-VL asymptomatic patients [84], further supporting a protective role for sCD40L in VL. sRAGE concentration was higher at EoT in Kenya, and had a tendency of increasing in the other countries at ToC. Serum sRAGE concentration increased during host response to infection [60], and might reflect overstimulation of membrane-bound RAGE, being a marker for inflammation [47].

We only observed three cases of VL relapse in our cohort, two in Ethiopia and one in Sudan. Both patients from Ethiopia had high parasite burden at admission, evidenced by a splenic aspirate grade 6+, in agreement with previous studies that reported parasite burden as a predictor of treatment failure in HIV and no-HIV coinfected VL patients [85,86]. Both patients from Ethiopia, but not the relapsed patient from Sudan had persistent splenomegaly at the end of treatment, which is previously reported as a marker for relapse risk in Sudan and India [30,31], and in VL-HIV co-infected patients from Ethiopia [87] and Brazil [88]. It is important to note that ~81% of the VL patients from Ethiopia had splenomegaly at EoT, and most did not relapse within 6 months of infection. Patient VL006 had the highest concentrations of sTNF.RI and II, and sCD25 after treatment. This is in agreement with previous data from Sudan, where patients that relapsed had higher concentrations of both sTNF RI and II at end of the treatment cycle, which were still high 4–6 months after treatment [72].

Splenomegaly, and to a lesser extent, hepatomegaly, are hallmarks of active visceral leishmaniasis (55). To investigate the association between 20 predictor variables and three clinical outcomes: hepatomegaly at admission, splenomegaly at EoT, and early markers predictive of persistent splenomegaly; we applied both PLS-DA and logistic regression (LR). PLS-DA was chosen for its robustness in handling multicollinearity and its capacity to identify predictive markers in multi-dimensional data settings (42, 85, 86). Logistic regression was employed as an interpretable

reference model, providing estimates of individual variable effects with adjustments for small and non-balanced sampling. Model building was challenging due to the limited sample size, imbalanced case/control distribution, and significant heterogeneity across patients. Hence, the identified markers are only exploratory and would benefit from future validation. The datasets from Ethiopia and Kenya each included approximately 30 samples with complete datasets, while Sudan and Uganda had around 15. As a result, reliable predictive modelling was only feasible for Ethiopia and Kenya. With Ethiopia and Kenya samples, both PLS-DA and LR approaches yielded consistent results, reinforcing the reliability of the identified predictive markers and the overall robustness of the classification framework. In line with the treatment response patterns observed, we identified sTNF receptors as potential markers for hepatomegaly in Ethiopia, and to persistent splenomegaly in Kenya. Elevated concentrations of sCD40L were identified as protective to prevent persistent splenomegaly in Ethiopia. These markers should however only be considered as indicators of the predicted outcomes and interpreted with caution.

This study has some limitations. As a multi-country project conducted in four EA nations, it faced logistical, data collection and standardization challenges. While patient retention was high, there were differences in data collection protocols between sites. For instance, there was no collection of post-treatment haematological and biochemical data from patients in Uganda. Another challenge was the low number of female patients and relapses, which precluded statistical support and conclusions of directly comparing these with their counterparts. Similarly, imbalanced group sizes affected the accuracy of prognostic models for splenomegaly and hepatomegaly. Despite these constraints, the combination of clinical/ haematological and immunological data from a large cohort of patients across EA provided valuable insights into patient status before and after treatment for VL.

The recovery of hematological parameters, such as albumin, hemoglobin, white blood cell (WBC) count, and platelet count; alongside a reduction in systemic inflammatory markers including sTNF-RI, sTNF-RII, sCD25, and sST2, and an increase in sCD40L, may be indicative of successful treatment of visceral leishmaniasis (VL). These biomarker dynamics suggest a partial resolution of inflammation and restoration of immune homeostasis during recovery. To confirm their predictive value, these markers should be evaluated in cohorts with a larger number of patients who experience treatment failure.

## Supporting information

**S1 Appendix. The Immstat@cure consortium members.**
(DOCX)

**S1 Supplementary Information. Protocol for the project.**
(PDF)

**S1 Strobe. Strobe checklist for cohort, case-control and cross-sectional studies was obtained from** https://www.strobe-statement.org/.
(DOCX)

**S1 Fig. Missing data across patients.** Each row corresponds to a clinical/Haematological/Immune marker, and each column corresponds to a patient. Red cells represent missing data. The colour-strip at the bottom represents the country of origin of each sample.
(DOCX)

**S2 Fig. Clinical data changes with VL patient treatment.** Violin plots showing the range and statistical support for the variations in clinical, haematological and inflammation markers pre and post treatment. The numbers of asterisks represent Wilcoxon signed-rank test (V1 x V2 comparison) or Mann-Whitney U test (HV comparing to V1 or V2) p-values, where 1–4 corresponds respectively values below 0,05, 0.01, 0.001 and 0.0001. A) Males Ethiopia, B) Males Kenya, C)

Males Sudan, D) Males Uganda, E) Females Kenya, F) Females Sudan, G) Females Uganda. Most of the after treatment clinical data for Uganda is missing. There is only one Female HV in Kenya.
(DOCX)

**S3 Fig. Slopegraph showing male patient's trait concentrations before (V1) and after (V2) treatment.** Each panel corresponds to a different trait. The normal reference ranges for the traits are represented by green (upper normal limit) and red (lower normal limit) dashed lines. Patients are identified by a combination of colour and shape. A) Males Ethiopia, B) Males Kenya, C) Males Sudan, D) Males Uganda.
(DOCX)

**S4 Fig. Slopegraph showing female patient's trait concentrations before (V1) and after (V2) treatment.** Each panel corresponds to a different trait. The normal reference ranges for the traits are represented by green (upper normal limit) and red (lower normal limit) lines. Patients are identified by a combination of colour and shape. A) Females Ethiopia, B) Females Kenya, C) Females Sudan, D) Females Uganda.
(DOCX)

**S5 Fig. Proportion of patients' clinical traits that were below (low) within range (green) or above the normal range for each assessed haematological marker.** Missing data is represented in gray (NA). Interval ranges for each patient group can be seen in S1 Table. Statistical comparisons between V1, V2 and HV can be seen in S2 Table. A) All data. B) Male patient data. C) Female patient data. D) Convenience controls data.
(DOCX)

**S6 Fig. Male-Female clinical, haematological and inflammatory markers comparison.** Violin plots showing the range and statistical support for the variations in clinical, haematological and inflammation markers in males (red) and females (blue). The numbers of asterisks represent Mann-Whitney U test p-values, where 1–4 corresponds respectively values below 0,05, 0.01, 0.001 and 0.0001.
(DOCX)

**S7 Fig. LEGENDplex standard curves.** Each panel represents a different marker, and each different plate used in the experiment is shown by different colour. The red line represents the Limit of Quantification (LOQ) used for each country, which was based on the higher LOQ for a given marker for all plates from a given country. A) Ethiopia, B) Kenya, C) Sudan, D) Uganda.
(DOCX)

**S8 Fig. PCA, Scree plot, loading plot and patient group biplot for each country.** A) Ethiopia. B) Kenya. C) Sudan. D) Uganda.
(DOCX)

**S9 Fig. Inflammation immune markers in females across the four countries.** The left panel is a PCA biplot of the inflammation markers values for each patient. Each patient is represented by a dot, coloured in yellow (V1) blue (V2) and red (HV). The arrows represent how much each immune marker contributes to the principal components. The middle panel is a radar plot, where each segment corresponds to the median of patients in a group (V1, V2 or HV), scaled to the highest median in the three groups (ex: if V1, V2 and HV were respectively 20, 10 and 5, they would be represented as 100, 50 and 25) of one inflammation marker. Markers that were significantly ($p < 0.05$) higher in V1 or V2 are respectively coloured in yellow or blue. Markers that were not significantly different are in black. Violin plots showing the range and statistical support for the variations in inflammation markers pre and post treatment. The numbers of asterisks represent p-values, where 1–4 corresponds respectively values below 0,05, 0.01, 0.001 and 0.0001. The letters correspond to the results from A) Ethiopia, B) Kenya, C) Sudan and D) Uganda. The statistical results and number of samples used in each

comparison can be seen in <u>S2 Table</u>. The PCA and UMAP were generated with samples that had the complete set of inflammatory markers.
(DOCX)

**S10 Fig. Slope graph showing male patient's trait concentrations before (V1) and after (V2) treatment.** Each panel corresponds to a different trait. Patients are identified by a combination of colour and shape. A) Males Ethiopia, B) Males Kenya, C) Males Sudan, D) Males Uganda.
(DOCX)

**S11 Fig. Association between variation in sCD40L and Platelet levels before and after treatment.** Each dot corresponds to one patient. The X and Y axis represent, respectively, the log-normalized scaled delta (V2 -V2) of sCD40L and Platelet levels. In A) Ethiopia; B) Kenya and C) Sudan. D) Table summarizing the linear regression and Pearson correlation results. Lm_AdjRsquared, Lm_pvalue and Lm_Estimate are respectively the adjusted R squared, p-value and estimate for the logistic regression results, while the Cor_pvalue and Cor_rho represent the p-value and rho from the spearman correlation. Linear regression analysis showed that changes in sCD40L were associated with changes in platelet concentrations for Ethiopia ($\beta = 0.85$, $R^2 = 0.30$, $p = 0.0006$) and Sudan ($\beta = 2,2$, $R^2 = 0.44$, $p = 0.007$), but not Kenya ($\beta = 0.26$, $R^2 = 0.03$, $p = 0.8$). The model explained ~30–44% of the variance in sCD40L concentrations in Ethiopia and Sudan, indicating that while platelets contributed substantially, they were not solely responsible for the variation in sCD40L.
(DOCX)

**S12 Fig. Correlation between clinical, haematological and inflammation markers.** The left panel shows only correlations with FDR adjusted p-values $< 0.05$. The right panel shows all of the identified correlations, irrespective of the p-values. A) Ethiopia pre-treatment; B) Ethiopia post-treatment; C) Kenya pre-treatment; D) Kenya post-treatment; E) Sudan pre-treatment; F) Sudan post-treatment; G) Uganda pre-treatment; H) Uganda pos-treatment.
(DOCX)

**S13 Fig. Logistic regression using Age as covariate.** Logistic Regression coefficient with 95% confidence intervals for the association of each marker to the evaluated trait, using Age as a covariate. Increases in values of markers that are above or below zero respectively increases or decreases the Odds Ratio of the trait. Values with p-value <0.05 are presented by triangles. A) Ethiopia hepatomegaly in pre-treatment patients; B) Ethiopia, prediction of persistent splenomegaly based on pre-treatment data; C) Kenya persistent splenomegaly post-treatment.
(DOCX)

**S14 Fig. Hepatomegaly and splenomegaly marker assessment.** The left panel corresponds to boxplots showing the logistic regression coefficient values of the 1000 stratified bootstrap replicates. The Y axis represents the markers, and the X axis the logistic regression coefficient. The middle panel corresponds to the result of the logistic regression bootstrap replicates. Each dot corresponds to the result of one of the 1000 replicates, where the X axis represents -log10(p-value) and the Y axis represents the logistic regression coefficient, as a representation of the Odds Ratio. In both left and middle panels, positive and negative values correspond respectively to increased or decreased odds of having the evaluated the clinical outcome. In the middle panel, the vertical and horizontal red lines correspond respectively to the p-value of 0.05 and to the logistic regression coefficient of zero. The right panel corresponds to violin plots comparing hepatomegaly or splenomegaly patients in red (1), with patients without those symptoms in blue (2), using Mann Whitney u test. A) Hepatomegaly in Ethiopia; B) pre-treatment markers prediction of late (post-treatment) splenomegaly in Ethiopia; C) persistent splenomegaly in Kenya. ns: non-significant.
(DOCX)

**S15 Fig. Hepatomegaly and splenomegaly marker assessment.** Top left panel: sPLS-DA loading plot, showing the loading weight of the Inflammation/Clinical markers influence on the assessed trait, ordered from bottom (highest impact) to top (lowest impact). Values in yellow and blue are respectively associated with the presence or absence of the trait. Bottom left panel: Variable Importance in Projection (VIP) of the evaluated Inflammation/Clinical markers for a given trait. Values higher than 1.3 are highlighted in red. Left middle panel: Logistic Regression coefficient with 95% confidence intervals for the association of each marker to the evaluated trait. Increases in values of markers that are above or below zero respectively increases or decreases the Odds Ratio of the trait. Values with p-value <0.05 are presented by triangles. Middle right panel: corresponds to the result of the logistic regression bootstrap replicates. The X axis represents the logistic regression coefficient, as a representation of the Odds Ratio. Positive and negative values correspond respectively to increased or decreased odds of having the evaluated the clinical outcome. Right panel: Model prediction scores results for the PLS-DA leave one patient out (LOPO) cross validation, using increasing numbers of markers, selected based on highest VIP scores. PPV: Positive Predictive Value; NPV: Negative Predictive Value. A) Ethiopia persistent splenomegaly; B) Early hepatomegaly in Sudan; C) predictive power of early markers to identify late splenomegaly in Kenya; D) Early splenomegaly in Sudan; E) Early splenomegaly in Uganda.
(DOCX)

**S1 Table. Patient's Clinical, haematological and inflammatory markers measurements.** Summary of the Males and females are represented in the first two sheets. The third sheet represents the individual anonymized patient data.
(XLSX)

**S2 Table. Statistical evaluation of the difference of haematological and inflammatory markers values among pre-treatment (V1), post-treatment (V2) patients and healthy volunteers (HV).** Each sheet represented a different combination of sex and country.
(XLSX)

**S3 Table. Assessment of the number of patients in V1, V2 and HV that were within the reference range for the clinical and haematological traits.** Sheets 1, to 4 represents respectively all patients, males, females and healthy volunteers.
(XLSX)

**S4 Table. Correlation between patient clinical, haematological and inflammatory markers, in V1 and V2.** Each sheet corresponds to a different country.
(XLSX)

**S5 Table. Summary of the diagnostic metrics performance of the PLS-DA, with LOPO cross validation.** Each sheet corresponds to a patient stage (as V1, and V2) and clinical outcome (as splenomegaly).
(XLSX)

**S6 Table. Statistical evaluation of the difference between patients with different clinical outcomes (as hepatomegaly and no hepatomegaly) in different clinical stages and countries.**
(XLSX)

## Acknowledgments

We thank all patients and their families for agreeing to participate and their collaboration. The authors thank all members of the field teams at the study sites of Chemolingot, Amudat, Doka and Gondar, including clinicians, nurses, laboratory technicians, and the local communities for their contributions to the study. We also acknowledge the Ministries of Health of Kenya and Sudan; the Ministry of Education in Ethiopia and the National Council for Science and Technology in Uganda

for their support. The authors thank the African Centre for Community Investment in Health (ACCIH), a partner to the Chemolingot Sub County Hospital, Kenya, and the County Government of Baringo for allowing them to undertake the study at the Chemolingot hospital. Disclaimer. This manuscript is published with permission from the Director of Kenya Medical Research Institute.

## Author contributions

**Conceptualization:** Peter O'Toole, Charles J.N. Lacey, Jane Mbui, Asrat M. Hailu, Paul M. Kaye, Margaret Mbuchi, Ahmed M Musa, Joseph Olobo.

**Data curation:** Ayenew Addisu, Alice Bayiyana, Joao Cunha, Daniel Matano, Brima M. Younis, Karen Hogg.

**Formal analysis:** Joao Cunha, Karen Hogg, Rebecca Wiggins.

**Funding acquisition:** Peter O'Toole, Charles J.N. Lacey, Jane Mbui, Asrat M. Hailu, Paul M. Kaye, Margaret Mbuchi, Ahmed M Musa, Joseph Olobo.

**Investigation:** Ayenew Addisu, Alice Bayiyana, Joao Cunha, Daniel Matano, Brima M. Younis, Karen Hogg, Rebecca Wiggins, Wilson Biwott, Finnley Osuna, Christine Ichugu, Ayalew Jejaw Zeleke, Eleni Ayele, James Obondo Sande, Eltahir A.G. Khalil, Hussam M.H. Ibrahim, Mahmoud A. Mahmoud, Ahmed I.B. Zakaria, Brenda Adiko, Peter O'Toole, Charles J.N. Lacey, Jane Mbui, Asrat M. Hailu, Paul M. Kaye, Margaret Mbuchi, Ahmed M Musa, Joseph Olobo.

**Methodology:** Joao Cunha, Karen Hogg, Peter O'Toole, Charles J.N. Lacey, Jane Mbui, Asrat M. Hailu, Paul M. Kaye, Margaret Mbuchi, Ahmed M Musa, Joseph Olobo.

**Project administration:** Rebecca Wiggins, Flavia D'Alessio.

**Resources:** Peter O'Toole, Charles J.N. Lacey, Jane Mbui, Asrat M. Hailu, Paul M. Kaye, Margaret Mbuchi, Ahmed M Musa, Joseph Olobo.

**Supervision:** Joao Cunha, Karen Hogg, Peter O'Toole, Charles J.N. Lacey, Jane Mbui, Asrat M. Hailu, Paul M. Kaye, Margaret Mbuchi, Ahmed M Musa.

**Validation:** Rebecca Wiggins, Paul M. Kaye.

**Visualization:** Joao Cunha.

**Writing – original draft:** Joao Cunha.

**Writing – review & editing:** Ayenew Addisu, Alice Bayiyana, Joao Cunha, Daniel Matano, Brima M. Younis, Karen Hogg, Rebecca Wiggins, Wilson Biwott, Finnley Osuna, Christine Ichugu, Ayalew Jejaw Zeleke, Eleni Ayele, James Obondo Sande, Eltahir A.G. Khalil, Hussam M.H. Ibrahim, Mahmoud A. Mahmoud, Ahmed I.B. Zakaria, Brenda Adiko, Peter O'Toole, Charles J.N. Lacey, Jane Mbui, Asrat M. Hailu, Paul M. Kaye, Margaret Mbuchi, Ahmed M Musa, Joseph Olobo.

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
