## [Decision Letter · Decision Letter 0]

23 Dec 2025

Response to Reviewers
Revised Manuscript with Track Changes
Manuscript

Shaden Kamhawi

co-Editor-in-Chief

Paul Brindley

co-Editor-in-Chief

**Journal Requirements:**

At this stage, the following Authors/Authors require contributions: Ayenew Addisu, Alice Bayiyana, Joao Cunha, Daniel Matano, Brima M. Younis, Karen Hogg, Rebecca Wiggins, Wilson Biwott, Finnley Osuna, Christine Ichugu, Ayalew Jejaw Zeleke, Eleni Ayele, Obondo James Sande, Eltahir A.G. Khalil, Hussam M.H. Ibrahim, Mahmoud A. Mahmoud, Ahmed I.B. Zakaria, Brenda Adiko, Peter O’Toole, Flavia D’Alessio, Charles J.N. Lacey, Jane Mbui, Asrat M. Hailu, Margaret Mbuchi, Paul M. Kaye, Ahmed M Musa, and Joseph Olobo. Please ensure that the full contributions of each author are acknowledged in the "Add/Edit/Remove Authors" section of our submission form.

3) Please amend your detailed Financial Disclosure statement. This is published with the article. It must therefore be completed in full sentences and contain the exact wording you wish to be published.

State what role the funders took in the study. If the funders had no role in your study, please state: "The funders had no role in study design, data collection and analysis, decision to publish, or preparation of the manuscript.".

**Reviewers' comments:**

**Key Review Criteria Required for Acceptance?**

**Methods:**

-Are the objectives of the study clearly articulated with a clear testable hypothesis stated?

-Is the study design appropriate to address the stated objectives?

-Is the population clearly described and appropriate for the hypothesis being tested?

-Is the sample size sufficient to ensure adequate power to address the hypothesis being tested?

-Were correct statistical analysis used to support conclusions?

-Are there concerns about ethical or regulatory requirements being met?

Reviewer #1: (No Response)

Reviewer #2: Yes

Reviewer #3: The study is well designed with proper statistical analysis.

**Results**

-Does the analysis presented match the analysis plan?

-Are the results clearly and completely presented?

-Are the figures (Tables, Images) of sufficient quality for clarity?

Reviewer #1: (No Response)

Reviewer #2: Yes

Reviewer #3: The analysis presented match the analysis plan. Data representation is clear. The bar graphs need higher resolution.

**Conclusions**

-Are the conclusions supported by the data presented?

-Are the limitations of analysis clearly described?

-Do the authors discuss how these data can be helpful to advance our understanding of the topic under study?

-Is public health relevance addressed?

Reviewer #1: (No Response)

Reviewer #2: Yes

Reviewer #3: Yes, the conclusions are supported by the data presented

**Editorial and Data Presentation Modifications?**

Reviewer #1: (No Response)

Reviewer #2: (No Response)

Reviewer #3: 1. The concern I have is that the rationale of studying those four populations is missing. Ethiopia, Kenya, Uganda, and Sudan are genetically diverse regions reflecting ancient migrations and linguistic groups (Afro-Asiatic, Nilo-Saharan, Bantu), with East Africa showing deep roots, high mtDNA and other autosomal haplogroup variability. I would suggest to include this point in the introduction section. The differences in outcome may stem from the differences in the genetic make-up of the unique populations studied.

2. Authors mentioned that Leishmaniasis patients exhibit co-infection with HIV. It would be useful to have some comparative information on inflammatory markers in VL and VL-HIV patients.

**Summary and General Comments**

Reviewer #1: (No Response)

Reviewer #2: In this manuscript, Addisu and coworkers investigate the clinical and immunological profiles of visceral leishmaniasis patients from four East African countries Ethiopia, Kenya, Sudan, and Uganda at admission and at the end of treatment with sodium stibogluconate and paromomycin. The study reports clear treatment associated improvements in clinical parameters and inflammatory markers. The authors also attempt to identify blood based markers that can predict either enlarged spleen or liver size. Overall, the study provides valuable descriptive data from four East African regions where VL burden is high. It documents consistent improvements in clinical and inflammatory markers at the end of treatment. However, the predictive components should be toned down and contextualized as exploratory rather than definitive. While the study is important, there are a few concerns that needs to be addressed by the author:

1. A significant limitation of the study, as acknowledged by the authors, is the relatively low number of samples per country. The sample size becomes even smaller when cases with incomplete data are excluded from downstream analyses. This limitation is further compounded by substantial heterogeneity in clinical and immunological profiles across the four regions, which necessitated analyzing each country separately. The authors should discuss this limitation more explicitly and support it with appropriate references to similar epidemiological or biomarker studies that faced comparable sample size constraints. Another important limitation is the skewed sex distribution in the cohort, which is predominantly male. The authors should address whether and how the findings can be generalized to females, and discuss any known sex related differences in VL presentation or immune responses.

2. The authors also attempt modelling approaches such as logistic regression and PLS DA to identify blood based markers predicting enlarged spleen or liver at baseline or incomplete recovery after treatment. However, because these analyses use only patients with complete data, the sample sizes become even smaller in some countries. The authors acknowledge the lack of robustness in predictive modelling under these constraints. Hence, these claims should be interpreted with caution.

3. In subsection 2.6 the authors state that analyses were performed in R but do not provide package versions, seeds, or code availability. Methods such as UMAP require reporting of parameters including seed, n neighbors, min dist, and metric because of their stochastic nature. Even PCA and variable selection criteria should be more clearly described. The authors should specify which clinical markers were included in PCA and UMAP, justify the choices, and clarify how missing Uganda V2 data were handled.

4. The interpretation that increased sCD40L at end of treatment indicates restored immune homeostasis is not fully supported. Because sCD40L levels closely track platelet activation and platelet abundance, the rise in sCD40L may simply reflect platelet recovery rather than improved anti leishmania immunity. Since platelet counts also increase in parallel especially in the Sudan cohort, normalizing sCD40L to platelet number would prevent over interpretation of platelet driven changes.

Minor Comments

1. Some modelling terms such as ‘dim1’ and ‘dim2’ can be unfamiliar to readers without a statistical background. Including brief definitions within the text or figure legends would help. Similarly, abbreviations such as SSG PM and HV should be defined at first use. For figures that include multiple immunological markers (for example, sTNFRI, sRAGE), the authors may include full marker names or short biological descriptors either in the figure legend or in a supplementary table for easier interpretation by a general audience.

2. The rational for choosing the inflammatory markers should be properly discussed in the introduction section citing appropriate references.

3. The treatment protocol (dose/duration etc.) should be clearly mentioned in the methods section.

4. The authors should also clarify whether a random sampling strategy was adopted, and if multiple participants were enrolled from the same household or family, they should discuss how potential hereditary biasness was avoided.

5. Axis labeling in most of the figures are too small and are difficult to read. The font size should be increased for better readability.

Reviewer #3: The manuscript 'Systemic inflammatory markers of visceral leishmaniasis treatment response in East Africa' is a comprehensive and well-designed study of the immune status of VL patients in East Africa comprising of Kenya, Uganda, Ethiopia and Sudan at the end of treatment. By integrating clinical findings with blood and inflammation markers, they have shown that clinical and inflammatory profiles varied at admission and reverted to healthy levels at treatment's end.

1. The concern I have is that the rationale of studying those four populations is missing. Ethiopia, Kenya, Uganda, and Sudan are genetically diverse regions reflecting ancient migrations and linguistic groups (Afro-Asiatic, Nilo-Saharan, Bantu), with East Africa showing deep roots, high mtDNA and other autosomal haplogroup variability. I would suggest to include this point in the introduction section. The differences in outcome may stem from the differences in the genetic make-up of the unique populations studied.

2. Authors mentioned that Leishmaniasis patients exhibit co-infection with HIV. It would be useful to have some comparative information on inflammatory markers in VL and VL-HIV patients.

3. The discussion is too lengthy. It needs to be succinct.

PLOS authors have the option to publish the peer review history of their article (what does this mean? ). If published, this will include your full peer review and any attached files.

**Do you want your identity to be public for this peer review?** For information about this choice, including consent withdrawal, please see our Privacy Policy .

Reviewer #1: No

Reviewer #2: No

Reviewer #3: **Yes:** Arnab Gupta

**Figure resubmission:**

**Reproducibility:** To enhance the reproducibility of your results, we recommend that authors of applicable studies deposit laboratory protocols in protocols.io, where a protocol can be assigned its own identifier (DOI) such that it can be cited independently in the future. Additionally, PLOS ONE offers an option to publish peer-reviewed clinical study protocols. Read more information on sharing protocols at https://plos.org/protocols?utm_medium=editorial-email&utm_source=authorletters&utm_campaign=protocols

---

## [Decision Letter · Decision Letter 1]

12 Feb 2026

Dear Prof. Kaye,

We are pleased to inform you that your manuscript 'Systemic inflammatory markers of visceral leishmaniasis treatment response in East Africa' has been provisionally accepted for publication in PLOS Neglected Tropical Diseases.

Best regards,

Syamal Roy

Academic Editor

Susan Madison-Antenucci

Section Editor

Shaden Kamhawi

co-Editor-in-Chief

Paul Brindley

co-Editor-in-Chief

Reviewer's Responses to Questions

**Key Review Criteria Required for Acceptance?**

**Methods**

-Are the objectives of the study clearly articulated with a clear testable hypothesis stated?

-Is the study design appropriate to address the stated objectives?

-Is the population clearly described and appropriate for the hypothesis being tested?

-Is the sample size sufficient to ensure adequate power to address the hypothesis being tested?

-Were correct statistical analysis used to support conclusions?

-Are there concerns about ethical or regulatory requirements being met?

Reviewer #1: (No Response)

Reviewer #2: Yes

Reviewer #3: Yes; yes; yes; yes; yes; yes

**Results**

-Does the analysis presented match the analysis plan?

-Are the results clearly and completely presented?

-Are the figures (Tables, Images) of sufficient quality for clarity?

Reviewer #1: (No Response)

Reviewer #2: Yes

Reviewer #3: Yes; Yes; Yes

**Conclusions**

-Are the conclusions supported by the data presented?

-Are the limitations of analysis clearly described?

-Do the authors discuss how these data can be helpful to advance our understanding of the topic under study?

-Is public health relevance addressed?

Reviewer #1: (No Response)

Reviewer #2: Yes

Reviewer #3: Yes; yes; to an extent; Yes

**Editorial and Data Presentation Modifications?**

Reviewer #1: (No Response)

Reviewer #2: NA

Reviewer #3: Accept

**Summary and General Comments**

Reviewer #1: (No Response)

Reviewer #2: All comments/concerns have been addressed. The manuscript may be accepted in its present form

Reviewer #3: Authors have satisfactorily responded to all my queries. The study is comprehensive and I strongly believe that this will add to the existing knowledge in the field of visceral Leishmaniasis primarily prevalent in regions of East Africa

PLOS authors have the option to publish the peer review history of their article (what does this mean? ). If published, this will include your full peer review and any attached files.

**Do you want your identity to be public for this peer review?** For information about this choice, including consent withdrawal, please see our Privacy Policy .

Reviewer #1: No

Reviewer #2: No

Reviewer #3: **Yes:** Arnab Gupta

---

## [Editor Report · Acceptance letter]

Dear Prof. Kaye,

We are delighted to inform you that your manuscript, "Systemic inflammatory markers of visceral leishmaniasis treatment response in East Africa," has been formally accepted for publication in PLOS Neglected Tropical Diseases.

Best regards,

Shaden Kamhawi

co-Editor-in-Chief

Paul Brindley

co-Editor-in-Chief
